# Long-term coding of personal and universal associations underlying the memory web in the human brain

Emanuela De Falco[1], Matias J. Ison[1,2,†], Itzhak Fried[3,4,5,6] & Rodrigo Quian Quiroga[1,2]

Neurons in the medial temporal lobe (MTL), a critical area for declarative memory, have been shown to change their tuning in associative learning tasks. Yet, it is unclear how durable these neuronal representations are and if they outlast the execution of the task. To address this issue, we studied the responses of MTL neurons in neurosurgical patients to known concepts (people and places). Using association scores provided by the patients and a web-based metric, here we show that whenever MTL neurons respond to more than one concept, these concepts are typically related. Furthermore, the degree of association between concepts could be successfully predicted based on the neurons' response patterns. These results provide evidence for a long-term involvement of MTL neurons in the representation of durable associations, a hallmark of human declarative memory.

[1] Centre for Systems Neuroscience, University of Leicester, Leicester LE1 7QR, UK. [2] Department of Engineering, University of Leicester, Leicester LE1 7RH, UK. [3] Department of Neurosurgery, David Geffen School of Medicine, University of California Los Angeles, Los Angeles, California 90095, USA. [4] Semel Institute for Neuroscience and Human Behavior, University of California Los Angeles, Los Angeles, California 90095, USA. [5] Sackler Faculty of Medicine, Tel-Aviv University, Tel-Aviv 69978, Israel. [6] Functional Neurosurgery Unit, Tel-Aviv Medical Center, Tel-Aviv 64239, Israel. † Present address: School of Psychology, University of Nottingham, University Park, Nottingham NG7 2RD, UK. Correspondence and requests for materials should be addressed to R.Q.Q. (email: rqqg1@le.ac.uk).

It has long been recognized that the hippocampus and its neighbouring structures in the medial temporal lobe (MTL) play an essential role in declarative memory[1–4], involving, in particular, the encoding of associations between items[3,5]. Studies in animals have shown the engagement of MTL neurons in associative learning[6–13]. Consistent with these findings, we have recently shown that MTL neurons in humans rapidly change their tuning to encode new associations[14]. However, in all of these studies, recordings were done while subjects performed associative learning tasks. Therefore, it is not clear whether the MTL provides a transient encoding during learning that is created afresh for each new memory and then consolidates in cortex[2], or a more stable representation that persists after task execution[1]. Evidence in favour of one or the other model has been based on lesion or imaging studies[15], but there is so far no direct evidence of neurons coding (or not) previously acquired and not task-related associations.

To address this issue, following a previous observation of neurons responding to well-known and allegedly associated concepts (for example, two co-stars in a television show)[16,17], we designed a systematic study to determine if these co-activations were just random coincidences or if there is a consistent tendency for MTL neurons to encode meaningful associations, independent of the execution of an associative learning task. For this, we evaluated the neurons' responses to presented images and, in 24 experimental sessions performed by 12 patients, we asked the subjects to rate how much they related a subset of 10–15 images (including those eliciting responses) with each other. Complementing these results, we then used a web-based metric of 'universal associations' to study an eventual encoding of associated items with a larger number of experimental sessions ($N = 99$) in 49 patients.

We found that MTL neurons tend to fire to associated concepts, an effect that cannot be attributed to visual similarity between the stimuli, familiarity effects, recall of associated items or broad semantic categorizations. Moreover, we show a non-topographically organized distribution of responses, which is ideal for the fast formation of new associations.

## Results

**Personal association metric.** In the 24 experimental sessions for which we obtained the patients' personal association scores, we found 32 units (19 single units and 13 multi-units) that responded to more than 1 of the presented pictures (mean: 3.1 pictures per neuron; s.d.: 4.8). The number of units used for this and the following analyses is displayed in Supplementary Table 1. Figure 1a shows one of these neurons, which significantly responded (see Methods) to the pictures of 3 relatives of the patient and 2 celebrities. On the basis of the patient's association scores (Fig. 1b) the relatives were clearly related to each other and the two celebrities were related to one of the relatives (stimulus 5). Note that the neuron did not fire to all family members (for example, it did not fire to stimulus 7) or celebrities, but only to a subset of the people that the patient considered to be related. From the patients' entries, for each unit we calculated a mean association score between the pairs of stimuli eliciting responses ($AS_{R-R}$) and between the pairs of stimuli, where one of them elicited a response but not the other one ($AS_{R-NR}$) (see Methods). In the example of Fig. 1, the mean association score between pairs of stimuli eliciting responses ($AS_{R-R} = 0.89$) was larger than the one for the other pairs ($AS_{R-NR} = 0.12$). In line with this observation, $AS_{R-R}$ values were significantly larger than $AS_{R-NR}$ ones when considering the population of 19 single units with more than 1 response ($P < 0.005$; Wilcoxon signed-rank test; $n = 19$). The same tendency was observed for the population of 13 multi-units, although in this case

the difference was not significant (which can be attributed to the fact that some of the response pairs may correspond to the firing of different neurons).

**Web-based association metric.** Given that it is not, in practice, possible to ask subjects to rate all of the few thousand pairwise associations between the about 100 images shown in an experimental session, we defined a web-based association score that we applied to a database of 99 experimental sessions (recorded in 49 patients), in which we had identified 261 units (129 single units and 132 multi-units; Supplementary Fig. 1) with more than one picture eliciting a response (mean: 3.0 pictures per neuron; s.d.: 3.9). The web-based score rates the degree of association between two images based on the number of 'hits' given by a joint search, normalized by the number of hits of the individual searches (see Methods). Figure 2a displays the matrix of web-association scores for the pictures used in all the experimental sessions, where we observe a clustering of values according to the broad categories of the images (actors, sportsmen and so on). The category labels shown in the figure were manually assigned for each stimulus, but a similar classification was also obtained using a clustering algorithm (see Methods; Supplementary Fig. 2). The scores shown in Fig. 2a reflect 'universal associations' between items, given by the shared inputs of billions of internet users, whereas the inputs by the patients reflect not just universal associations but also relationships given by the subject's preferences and personal experiences. Figure 2b shows the correlation between the personal and the web-based association scores (see Methods). As expected, we observe a strong correlation for highly associated items (for example, Bill Clinton and Hilary Clinton are associated both for the patients and for the web) and more variability for those that are less associated. The personal and web-association scores were significantly correlated for 11 out of 12 subjects (rank test with $P < 0.05$; 1,000 surrogates; see Methods) and, considering the data of all subjects, the correlation between both association scores was significantly higher than chance ($P < 10^{-3}$; Wilcoxon signed-rank test; $n = 12$; see Methods). On the basis of these results, we took the universal web-association scores as a proxy for the personal scores by the patients, which, in spite of individual differences, gave a reasonable approximation on average.

Using the web-based scores with the 261 units (recorded in 49 patients) with more than one response, we also found that pairs of images for which the neuron fired ($A_{R-R}$) were significantly more associated than other pairs of images ($A_{R-NR}$; Fig. 3), both for the single units ($P < 10^{-7}$; Wilcoxon signed-rank test; $n = 129$) and the multi-units ($P < 10^{-5}$; Wilcoxon signed-rank test; $n = 132$). Performing a visual similarity analysis, we found that this effect cannot be trivially attributed to the perceptual similarity between the images: three different visual similarity metrics gave significantly lower differences between the R-R and the R-NR values compared with the ones obtained with the web-based association scores (see Methods). This is in agreement with the fact that MTL neurons do not represent visual features, as they show visual invariance (that is, they respond to completely different pictures of the same person)[16] and their firing can be triggered by different sensory modalities (responding to the person's written and spoken name)[17]. Another potential confound is given by the familiarity of the images: since familiar pictures are represented by more neurons[18], there is in principle a higher chance of finding neurons firing to pairs of familiar pictures compared with non-familiar ones. To rule out this possibility, we performed two control analyses (see Methods: 'Effect of the relative number of hits'). First, we found that 'joint familiarity scores' (equation (6))

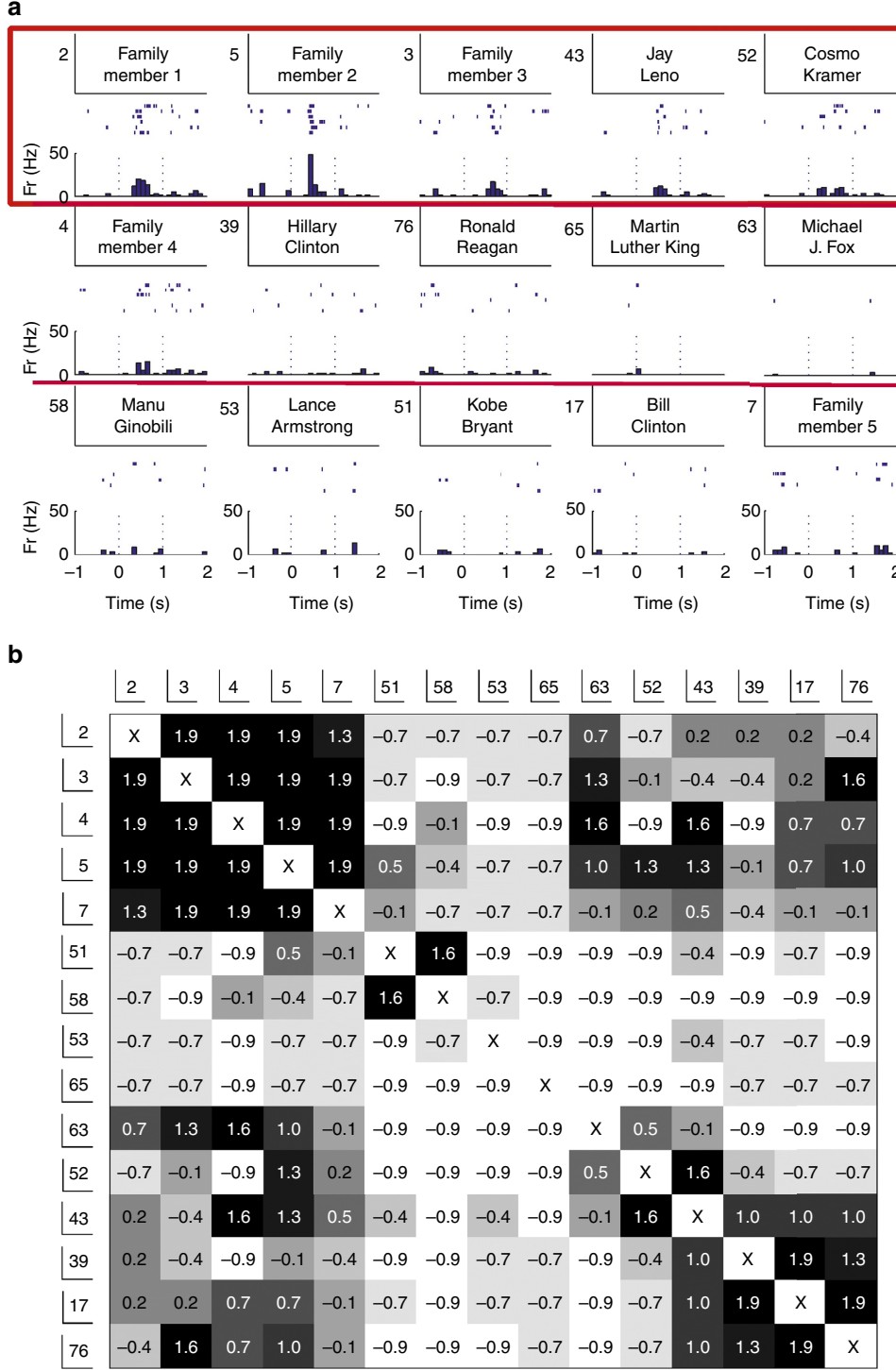

**Figure 1 | Exemplary neuron encoding long-term associations. (a)** A single neuron in the hippocampus that fired to pictures of the patient's relatives (pictures covered for confidentiality reasons) and to two celebrities that were related, according to the patient's report, to one of the relatives (stimulus 5). The red frame marks significant responses. Time zero marks the onset of picture presentations, shown for 1 s. **(b)** Personal association matrix filled by the patient with a *z*-score normalization. The association scores for the pictures to which the neuron fired (mean: 0.889) were higher than the scores obtained for other pictures (mean: 0.122). Owing to copyright issues, the images depicting people are replaced by their names. For the original figures see https://www2.le.ac.uk/centres/csn/publications-1/2016/longtermcoding.

for pairs of pictures to which the neurons fired were not statistically different than the ones for the other pairs. Second, we divided the image set into highly familiar and less familiar pictures, and found that results were statistically the same when considering association scores for pairs of responses where both pictures were highly familiar, where both pictures were less familiar and where one was highly familiar and the other was not (see Methods).

In addition, these results cannot attributed to a broad encoding of semantic categories, as $A_{R\text{-}R}$ versus $A_{R\text{-}NR}$ differences remained

**a**

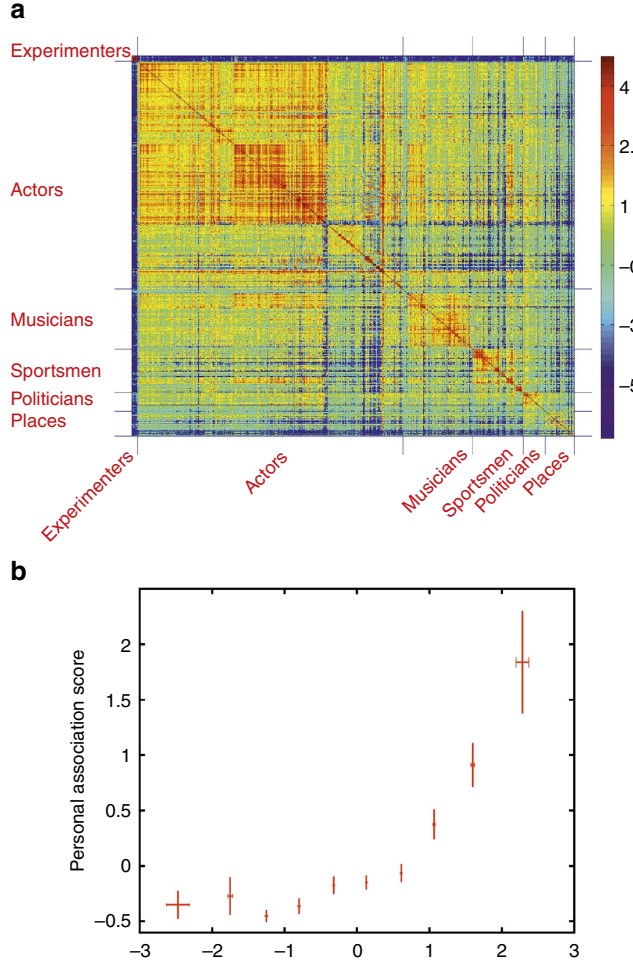

**b**

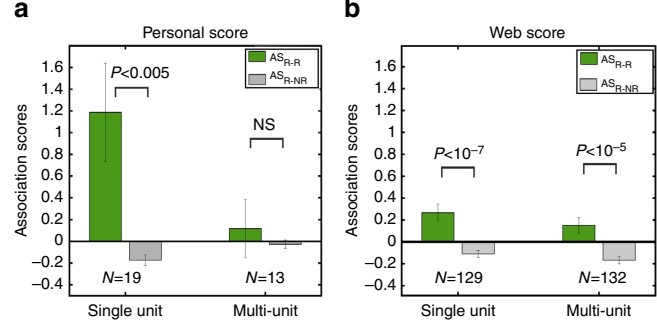

**Figure 3 | Mean association values with personal and web-based scores.** (**a**) Mean association score for the pairs of pictures to which the neuron fired (AS$_{R-R}$; green) and for the other pictures pairs (AS$_{R-NR}$; grey) based on the patients' scores. (**b**) Same as **a** but considering a larger set of responses using the web-based association scores.

**Figure 2 | Web-association matrix and correlation with personal scores.** (**a**) Web-association values between the 611 stimuli presented in the screening sessions. Note the clustering of values according to the broad categories of the images (manually assigned categories are shown on the left/bottom). The colour bar on the right shows the strength of association values in arbitrary units. (**b**) Correlation between the personal and the web-based association scores (see text for details; mean ± s.e.m.).

significant when constraining the comparisons to be within the same category (when the items eliciting responses belonged to the same category) or across the same categories (when they belonged to different categories) (see Methods). In other words, MTL neurons showed specific encoding of associations within and across semantic categories. Furthermore, for single units the difference between the web-association values for the pairs of images to which the neurons responded and the values for the other pairs (AS$_{R-R}$–AS$_{R-NR}$) (first two columns in Fig. 3b) were significantly lower than the ones obtained with the personal scores (first two columns of Fig. 3a) ($P < 0.05$; Wilcoxon rank-sum test; $n = 129$ and 19, respectively—the difference was not significant for the multi-units), in line with the fact that the web-based scores reflect universal associations and not the unique relationships made by each subject.

**Response latencies.** It is in principle possible that the responses to associated pictures are due to the fact that the neurons encode only one of the pictures, and the responses to the other ones are

due to cue-recall—that is, the associated picture acts as a cue to evoke the one encoded by the neuron, thus making it fire. Note, however, that the subjects saw about 100 images per session without doing any associative learning task. So, with such large number of images it seems unlikely that the subjects will spontaneously and consistently recall specific relationships that could explain the responses to associated items. To give further support, we estimated the latency of the responses using the same latency estimation method of Kreiman et al.[19] (see Methods), who studied human MTL single-neuron responses in a cue-recall paradigm. They report that the mean latency of human MTL responses obtained by evoking an image from a cue was of 409 ms (s.d.: 291 ms)—about 130 ms longer than the one they found for the visual responses (triggered by the picture presentations). In our case the mean latency of the responses was 253 ms (s.d.: 129 ms), which is well below the one reported by Kreiman et al. for cue-recall responses. Moreover, the mean latency difference of the responses to different stimuli (in the same neurons) was of 78 ms (s.d.: 89 ms), whereas the latency difference reported by Kreiman et al. for recall responses was more than 60% larger.

**Topographic organization.** Using similar calculations, we also studied whether there is a topographic organization of responses in the MTL, namely, that nearby neurons tend to respond to associated concepts. For this, we focused on the responses from 72 electrodes that had more than one unit (single- or multi-unit) separated after spike sorting, with at least one significant response each. We then quantified the degree of association between these pairs of responses and compared them with the ones for the other pairs. In line with previous evidence from studies in the rodent hippocampus[20], as well as from illustrative cases showing that neighbouring human MTL neurons tend to fire to completely unrelated things[21], the mean association score between the stimuli eliciting responses in these close-by units was not significantly different from the one for the other stimuli pairs (Wilcoxon signed-rank test; Fig. 4a), both when considering only the single units ($N = 49$) or all units together ($N = 159$), thus arguing against a topographic organization of responses.

More broadly, to compare the tendency of neurons to fire to associated concepts across the different MTL areas, we pulled together all units (given that results with the web-based scores were similar for single- and multi-units). For all areas, the association scores for the pairs of responsive stimuli AS$_{R-R}$ were significantly larger than the ones to other pairs (AS$_{R-NR}$) (in all cases $P < 0.05$; Wilcoxon signed-

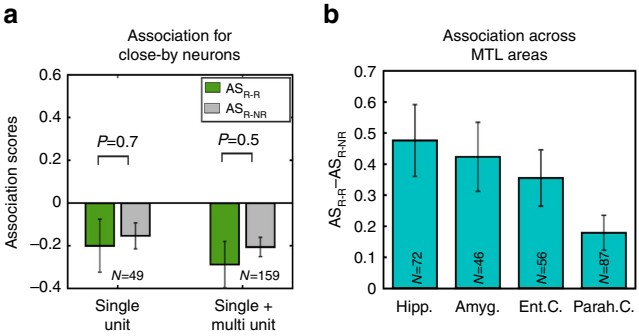

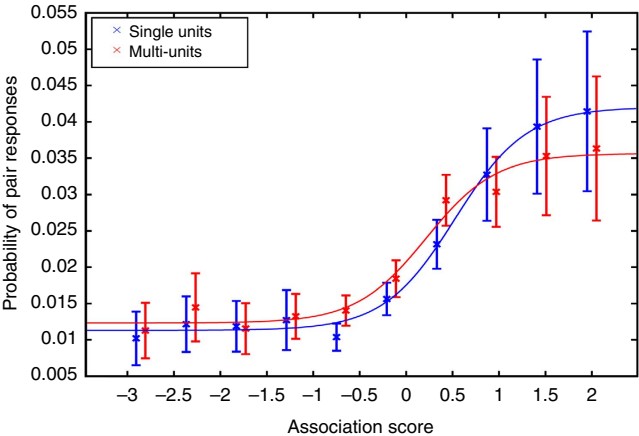

**Figure 4 | Topographic analysis. (a)** Mean association scores $AS_{R-R}$ (green) and $AS_{R-NR}$ (grey) between pictures eliciting responses in nearby neurons, which were recorded from the same electrode and separated after spike sorting. In this case $N$ indicates the number of electrodes with at least two responsive units separated after spike sorting. (**b**) Mean association score for the pairs of images eliciting responses for the different MTL areas.

**Figure 5 | Probability of responses to associated concepts.** Probability of responses to a pair of pictures as a function of their degree of association (see text for details). Error bars show s.e.m.

rank test; for sample sizes see Supplementary Table 2). The difference between association scores for response pairs and other pairs ($AS_{R-R}$–$AS_{R-NR}$) across MTL areas showed a tendency, being largest in the hippocampus and lowest in the parahippocampal cortex (Fig. 4b), which, however, did not reach statistical significance ($P = 0.07$; analysis of variance, $n = 129$ and 132).

**Probability of responses to associated concepts**. With the web-based scores, we also assessed how the probability of neurons responding to a pair of items depends on the degree of association between the items. That means, given a neuron that responds to a picture, we quantified the probability that the neuron responds to a second picture as a function on its association with the first one (see Methods). Figure 5 shows the population average, where we observe a psychometric-like nonlinear increase of the probability of pair responses with the degree of association between the items. In particular, the probability of neurons responding to highly associated items saturated at about 4%, going down to about 1% for weakly associated ones. This difference might, however, be larger considering that certain items, which were not associated according to the web metric, may have had a particular relationship for the patient, whereas there is more consistency between the subjective and web-metric scores for the highly associated items (Fig. 2b).

**Cell-by-cell and decoding analysis**. For each responsive neuron we performed a cell-by-cell analysis by correlating the matrix of 'joint-neural responses'—that is for each pair of pictures ($i,j$), the product of the responses to pictures $i$ and $j$—with the matrix of web-association values. To illustrate this, Fig. 6a shows a neuron that fired to a few basketball players and to other sportsmen. Figure 6b,c shows the corresponding joint-neural and web-association matrices between all 86 stimuli shown in this experimental session. We observe a clustering in the joint-neural responses, which partially matches the clustering present in the matrix of web-association scores. As described before—that is, consistent with the fact that the maximum probability of pair responses is about 4%—the neural responses encode relatively few of the associations between the stimuli. This partial matching was, however, enough to give a significant correlation between the neural and the web-association matrices, as assessed with a permutation test (Fig. 6d; see Methods). Figure 7 shows another example of a neuron firing to two researchers performing experiments with the patient, which are very much related

according to the web-based metric. Note that the neuron did not fire to a third experimenter (stimulus 3) that was also related to the first two. As in the previous case, the correlation between web-association and joint-neural matrices was significant. More examples are shown in the Supplementary Information (Supplementary Figs 4–7). Repeating this analysis with the 399 responsive units (174 single- and 225 multi-units) with a non-zero joint-neural response matrix (Supplementary Table 1), we obtained a significant correlation between the neural and web-association matrices in 19% of the cases (38 single- and 38 multi-units; rank test with $P < 0.05$; 1,000 surrogates). This relatively low number of significant correlations is, again, due to the fact that neurons encode relatively few of the associations. For example, the neuron in Fig. 8 responded to two clearly associated concepts (Superman and Mr. Incredible), but the correlation between the joint-neural and the web-association matrices was not significant due to the sparseness of the joint-neural matrix (see Supplementary Fig. 7 for a similar example). However, considering the population of 399 neurons, the correlation values were significantly larger than chance ($P < 10^{-6}$ Wilcoxon signed-rank test; $n = 399$). In line with this result, the correlation between the average joint-neural and web-association matrices obtained when considering all responses and pictures (that is, the matrices of Supplementary Fig. 3 and Fig. 2a, respectively) was also significantly larger than chance ($P < 0.01$, rank test, 1,000 surrogates).

We also used the neural responses to predict the degree of association between items (see Methods). We applied this analysis to 345 units (136 single- and 209 multi-units) with at least two non-zero entries in the joint-response matrix (Supplementary Table 1) and found that predictions were significantly larger than chance for 62 (18%) of these units (27 single- and 35 multi-units), using a rank test with $P < 0.05$ and 1,000 surrogates. The reason for this relatively low rate of success is, again, due to the fact that MTL neurons encode relatively few of the associations between the stimuli (see examples in Fig. 8 and Supplementary Fig. 7). In line with this observation, we found a significant correlation ($P < 0.005$; Spearman rank correlation test; $n = 345$) between the prediction error obtained for each neuron (minus the prediction error for the 5-percentile of the surrogates; see Methods) and the number of non-zero responses (which gives the number of non-zero entries in the joint-neural response matrix and eventually the number of associations coded by the neuron).

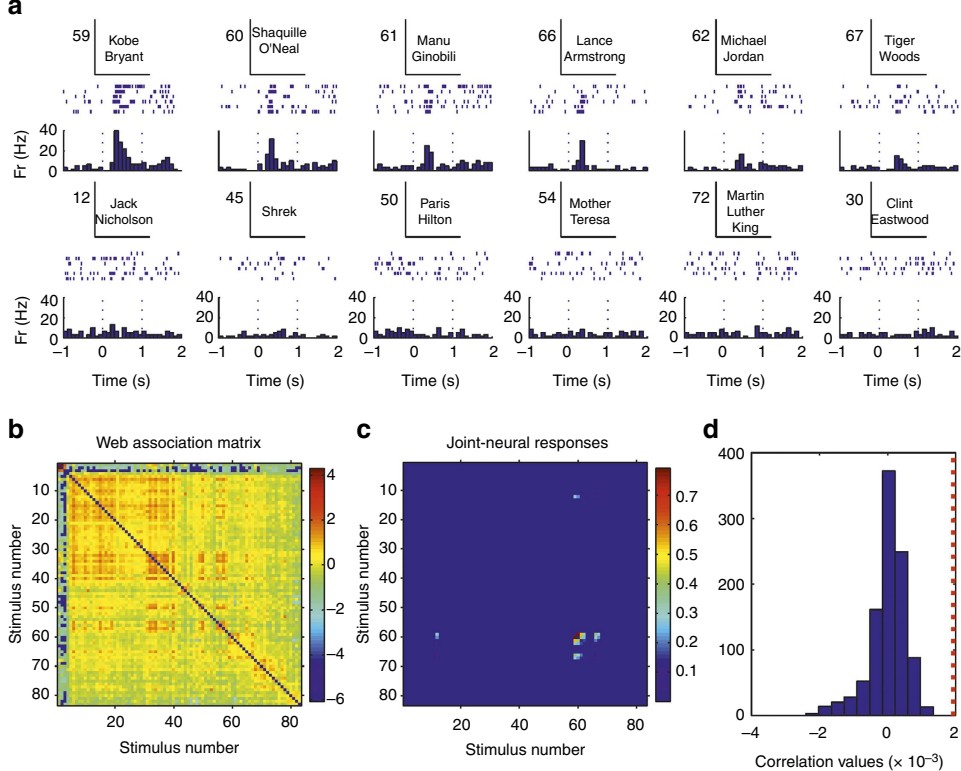

**Figure 6 | Correlation analysis with a neuron in the hippocampus. (a**) A single neuron in the hippocampus that fired to pictures of basketball players and to other sportsman. (**b,c**) Web-association and joint-neural response matrices (see text for details). Colour bars on the right denote association and join-response strength, respectively, in arbitrary units. (**d**) Correlation between the web-association and joint-response matrices (red dotted line) and for a distribution of 1,000 surrogates, which were obtained by randomly shuffling the responses (see text). The original correlation value was significantly larger than chance (rank test compared with the population of the 1,000 surrogates) with $P < 0.005$.

Altogether, considering the whole population of neurons, predictions were larger than chance with $P < 0.05$ (Wilcoxon signed-rank test; $n = 345$).

## Discussion

By analysing the association scores given by the subjects and by using a web-based association metric with a much larger database of responses, we demonstrated that whenever MTL neurons fire to more than one concept, these concepts tend to be associated. This claim is supported by three different analyses: first, the association scores for pairs of pictures to which neurons fired were significantly larger than the ones for other pairs of pictures. Second, a cell-by-cell analysis showed a correlation above chance between the neural responses and the association matrices (without using a criterion to define a response). Third, based on the neuron's responses we could predict the degree of association between pictures significantly better than chance. Moreover, we have shown that these results could not be trivially attributed to an effect given by the visual similarity of the images, their relative familiarity or the recall of a single picture triggered by associated ones.

Following on from these results, we also showed that the responses to associated items were more correlated to the subjects' scores, determined by their own personal experiences, than to 'universal' web-association values. In line, the neurons' responses reflected specific relationships between individual items and cannot be merely attributed to broad semantic categories—which is in contrast to findings in the monkey hippocampus[22], although in this case, category responses were argued to arise to optimally perform a delay match to sample task involving large number of images. In fact, MTL neurons tended to fire to some,

but not all of the pictures in a category—for example, the neuron shown in Fig. 1 fired only to some of the family members presented; the neuron in Fig. 7 fired to two of the experimenters, but not to another one and so on. To quantify this observation at the population level, we showed that first, when the items that a given MTL neuron fired to belonged to the same category (for example, two actors), the association scores for these items were, on average, significantly larger than the ones for other items in the same category (other actors); second, MTL neurons also responded to pairs of pictures belonging to different semantic categories (for example, an actor and a place) and in this case the association scores for the specific items the neuron fired to were, on average, significantly larger than the association scores for other items across the same categories (other actors and places).

In terms of regional organization, there was a close to significant statistical tendency for a non-homogeneous coding of associations within the MTL areas, being largest in the hippocampus and lowest in the parahippocampal cortex. Furthermore, as reported in the rodent hippocampus[20], and in contrast to findings in (monkey) cortical areas[23], close-by neurons did not fire to related items. Such non-topographically organized representation is indeed ideal for the fast creation of associations between arbitrary (that is, not related) concepts[21]. In line with this observation, we have recently shown that human MTL neurons can rapidly encode newly learned associations between (at first) unrelated items[14] and another recent work[24] has reported that MTL neurons encode associations between contiguous items shown in a sequence.

We have previously postulated that associations in the MTL are encoded via partially overlapping assemblies; this means that a

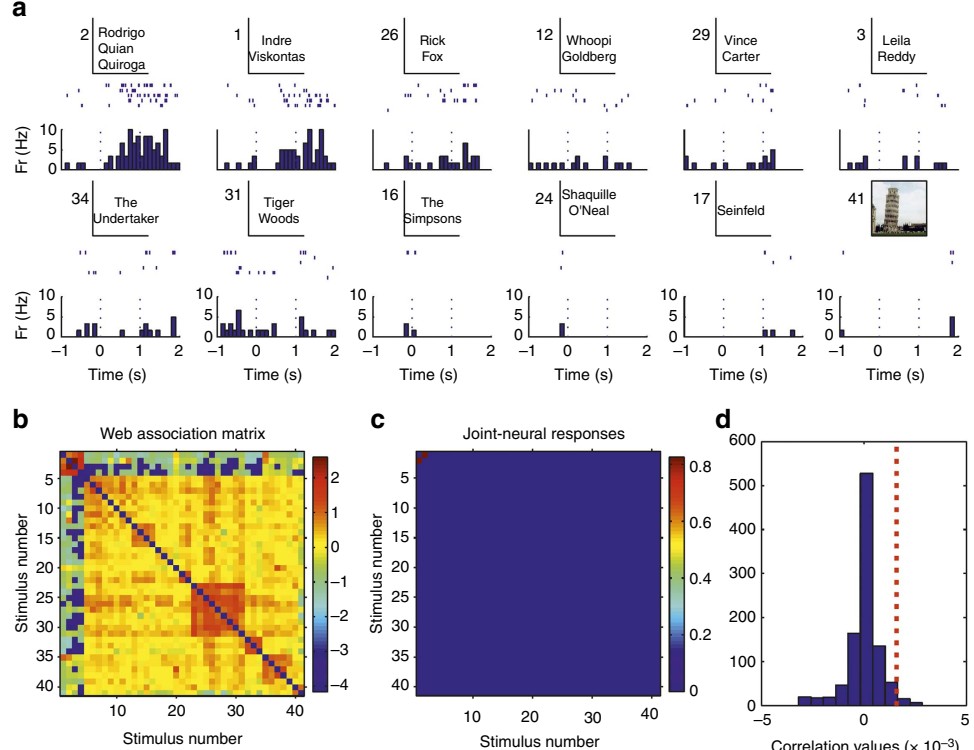

**Figure 7 | Correlation analysis with a neuron in the parahippocampal cortex.** (**a**) A single neuron in the parahippocampal cortex that fired to pictures of two researchers performing the experiments with the patient, but not to the picture of a third researcher also working with the patient (stimulus 3). (**b,c**) Web-association and joint-neural response matrices (see text for details). Colour bars on the right denote association and joint-response strength, respectively, in arbitrary units. (**d**) Correlation between the web-association and joint-response matrices (red dotted line) and for a distribution of 1,000 surrogates. In this case the correlation between the web-association and joint-response matrices was significantly larger than chance ($P < 0.05$, rank test, 1,000 surrogates).

proportion of neurons firing to a certain concept may also fire to related ones[21]. This way, the neural representation corresponding to different instances of the same concept will have a relatively large overlap—for example, different pictures of Luke Skywalker, as well as his written or spoken name, will trigger the firing of a similar set of neurons—whereas the neural representation of associated concepts will have a much lower overlap, as the Luke Skywalker assembly may trigger the firing of the one representing Yoda, but also the one representing Darth Vader, Han Solo and so on. With this model in mind, an intriguing finding from our previous study was the relatively large proportion of neurons (about 40%) that encoded newly learned associations[14]. In this respect, we now show that the long-term probability of joint responses to picture pairs is an order of magnitude lower (about 4% for highly related concepts and <1% for non-related ones). Therefore, we can argue that from a relatively high proportion of neurons initially encoding new associations, only a small fraction of these will consolidate this information into long-lasting representations in the MTL.

There is a long ongoing debate about the specific role of the MTL in the consolidation of declarative memory. One view, the standard consolidation model[2], is that the MTL has only a temporary role during learning and then memories consolidate into stable representations in cortex. An alternative view, the memory trace theory[1], is that the MTL provides a long-lasting representation that continues to play a critical role for declarative (and particularly episodic) memory after learning. Evidence in support of one or the other model has been largely based on lesion studies and the investigation of human amnesic patients[2,15,25–29], and has provided mixed results, which can be

attributed to the varying degrees of the lesions and the presence of compensatory mechanisms that can be used to perform the behaviour at test. Direct evidence from neuron's recordings is scarce. In particular, about 15% of place cells in the rodent hippocampus have been shown to maintain their tuning for up to 30 days in a familiar environment[30]. This result supports the idea of a long-term coding in the hippocampus, but it is an open issue whether, and to what extent, rodent place cells can be taken as a model of declarative memory. Miyashita and colleagues used a pair learning task and showed that perirhinal cortex activations preceded the ones in visual cortical areas, thus suggesting a long-term involvement of the MTL in encoding associations and giving feedback to visual areas[31], in line with their previous finding of a disruption of pair coding neurons in visual cortical areas on lesions in the entorhinal and perirhinal cortices[32]. Closer to our study, selective responses to well-learned associations were described in the monkey[11] and the rat hippocampus[12]. Moreover, using an eye-blink conditioning paradigm, neurons in the rabbit CA1 were shown to be active at the recall of remotely acquired associations[13]. In line with these findings, we have shown, in humans, the presence of a long-term coding of associations between known concepts. Note that our results do not rule out the possibility that these type of associations (beyond semantic category relationships) might be also encoded in cortex, but the fact that MTL neurons code previously existing, and not task related, associations—that is, independent of the specific task performed by the subjects—shows a long-term representation that goes beyond a temporary and malleable coding and offers new insights to our understanding of the role of the MTL in memory coding, its stability and capacity.

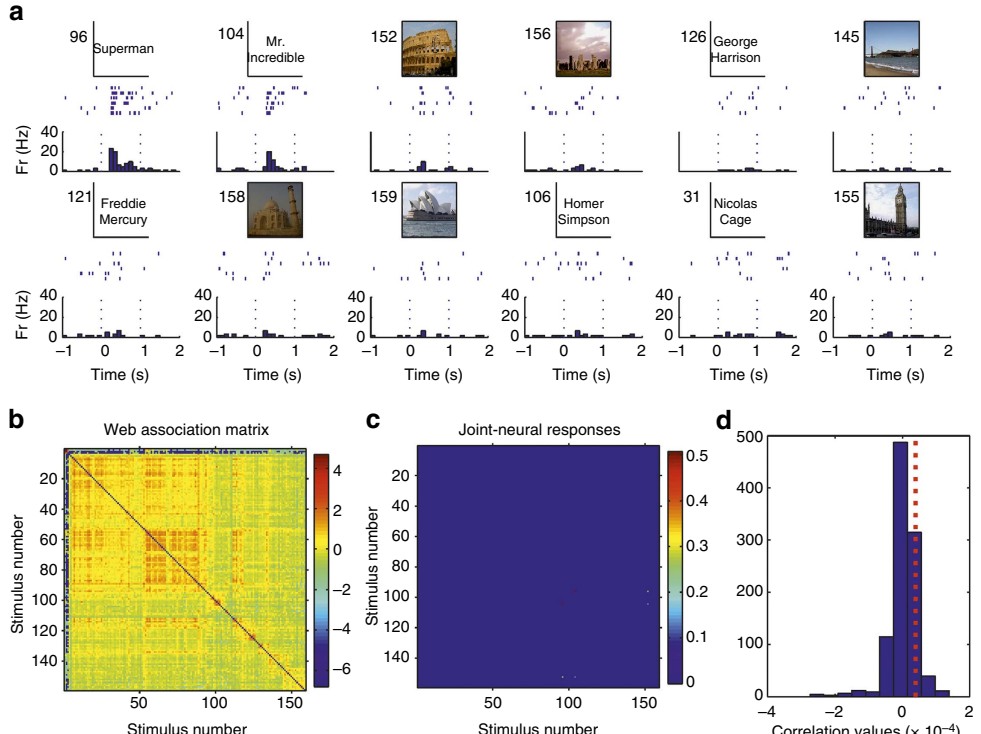

**Figure 8 | Correlation analysis with a neuron in the entorhinal cortex.** (**a**) A single neuron in the entorhinal cortex that fired to pictures of two cartoons, Superman and Mr. Incredible, but not to other cartoons, like Homer Simpson (stimulus 106). (**b,c**) Web-association and joint-neural response matrices (see text for details). Colour bars on the right denote association and join-response strength, respectively, in arbitrary units. (**d**) Correlation between the web-association and joint-response matrices (red dotted line) and for a distribution of 1,000 surrogates. In this case, the correlation between the web-association and the joint-response matrices was not significant ($P = 0.13$, rank test, 1,000 surrogates) due to the fact that the neuron encoded relatively few of the associations in the web-association matrix.

## Methods

**Subjects and recordings.** The data come from 99 experimental sessions in 49 patients with pharmacologically intractable epilepsy. Extensive non-invasive monitoring did not yield concordant data corresponding to a single resectable epileptic focus. Therefore, the patients were implanted with chronic depth electrodes for 7–10 days to determine the seizure focus for possible surgical resection[33]. Here we report data from sites in the hippocampus, amygdala, entorhinal cortex and parahippocampal cortex. All studies conformed to the guidelines of the Medical Institutional Review Board at University of California, Los Angeles, and all patients provided signed consent forms. The electrode locations were based exclusively on clinical criteria and were verified by CT co-registered to preoperative magnetic resonance imaging. Each electrode probe had a total of nine micro-wires at its end, eight active recording channels and one reference. For the first 41 patients (84 sessions), the differential signal from the micro-wires was amplified using a 64-channel Neuralynx system, filtered between 1 and 9,000 Hz and sampled at 28 kHz. For the remaining patients, a 128-channel Blackrock system was used using a filter between 0.3 and 7,500 Hz and a sampling frequency of 30 kHz.

**Experimental paradigm.** Subjects sat in bed, facing a laptop computer on which about 100 different pictures of known people and places (for example, actors, politicians and landmarks) were shown for 1 s, six times each in pseudo-random order. These pictures were partially chosen according to the subjects' interests and preferences. To check that the subjects were paying attention, after each presentation they had to respond whether the picture corresponded to a person or not. Each recording session lasted about 30 min. After the recordings, in a subset of 24 (out of the 99) sessions in 12 (of the 49) patients, subjects were asked to fill a 'personal association matrix', in which they ranked between 0 and 10 how much a subset of between 10 and 15 pictures were related to each other (between 45 and 105 comparisons in total). Entries given by the subjects were normalized with a $z$-score. The subset of the stimuli comprised those images eliciting responses in the recorded neurons, as well as other pictures presented in the experiments.

**Spike sorting and responsiveness criteria.** From the continuous wide-band data, spike detection and sorting was carried out using 'Wave_Clus', an adaptive and stochastic clustering algorithm[34]. The mean number of detected neurons per micro-wire was 1.6 (s.d.: 0.8). Neurons were classified into single- or multi-units

based on the following: (1) the spike shape and its variance; (2) the ratio between the spike peak value and the noise level; (3) the interspike interval (ISI) distribution of each cluster; and (4) the presence of a refractory period for single units; that is, < 1% spikes within < 3 ms ISI[17]. For each stimulus we defined a 'response window' between 200 and 1,000 ms, and a corresponding 'baseline window' between −1,000 and −200 ms. As in previous works[17], a response to a picture was considered significant if it fulfilled the following criteria: (i) the median number of spikes in the response interval was higher than the average baseline plus 5 s.d.'s; (ii) the median number of spikes in the response window was at least 2; (iii) the number of spikes in the response period was significantly higher than the one in baseline, according to a paired $t$-test ($P$ value < 0.05). For each significant response, the response latency was estimated from the spike density function (s.d.f.), as in previous works[19]. The s.d.f. was obtained convolving the spike train with a Gaussian of 100 ms width and then averaging across trials. The latency was then computed as the time where the s.d.f. crossed the baseline plus 2 s.d. value for at least 50 ms.

**Web-based association metric.** The matrices of associations filled by the patients included only a small subset (of the order of hundreds) of all possible association pairs (of the order of thousands) between the images shown in each recording session. Then, to automatically estimate the degree of relationship between the concepts presented in each recording session (and to include sessions where we did not have personal scores by the patients) we used an internet search engine (BING) and compared the number of hits to the joint searches with the number of hits to the individual searches. This type of metric is based on the pointwise mutual information[35], which has been used, for example, in linguistics to search for words synonyms[36]. It relies on the idea that the name of associated concepts will often appear together in web pages. The association score for each picture pair was defined as

$$a_{ij} = \log_2 \left( \frac{\text{hits}(\text{concept}_i \text{ AND } \text{concept}_j)}{\text{hits}(\text{concept}_i) \cdot \text{hits}(\text{concept}_j)} \right) \quad (1)$$

where the AND operator used in an internet search gives the number of pages containing both concepts. We limited the web search to famous people and places—that is, those concepts that are 'searchable' on the web and give a reasonable number of hits to have a reliable statistic (excluding names of family

members). As with the personal matrices, the values for each recording session were normalized using a z-score.

The stimuli were manually assigned to broad semantic categories (actors, sportsmen and so on; Fig. 2a). This manual classification had an 84% overlap with the classification obtained using a clustering analysis (Louvain community detection algorithm[37,38]), considering the same number of categories.

**Correlation between personal and web-association metrics.** The normalized association scores given by the subjects and the ones obtained with the web searches for the same pairs of items (again, excluding in both cases family members) were compared using the scalar product between both association scores. To assess statistical significance, for each subject we statistically compared, using a rank test with $P < 0.05$, the original correlation value with the ones obtained from a distribution 1,000 surrogates, created by randomly permuting the index of one of the association scores. At the population level we compared the original correlation values with the median of the surrogates for each subject using a right-sided (that is, our hypothesis is that the original values are higher than the ones of the surrogates) Wilcoxon signed-rank test. Moreover, to illustrate the correlation between the personal and web-based scores, we considered the web-based association scores for which we had the personal score given by the patient; then we binned these web-based scores into 10 equally spaced intervals, and plotted the mean ($\pm$ s.e.m.) value in each bin (x axis) against the mean ($\pm$ s.e.m.) value of the corresponding personal score (y axis) (Fig. 2b).

**Mean association scores.** For each neuron with a significant response to two or more images we defined a mean association score between the pair of images eliciting responses ($AS_{R-R}$), and a mean association score between the pair of images, where one elicited a response and the other did not ($AS_{R-NR}$). In other words, for each responsive picture we compared its association scores with the other pictures eliciting responses to the ones with the other pictures not eliciting responses. More specifically, denoting by N the number of pictures presented in a session and by k the number of responsive stimuli, we defined the mean association score $AS_{R-R}$ as

$$AS_{R-R} = \frac{1}{\binom{k}{2}} \sum_{i=1}^{k-1} \sum_{j=i+1}^{k} a_{ij} \tag{2}$$

where $a_{ij}$ are the z-score-normalized association scores from the personal matrices (or from equation (1) for the web-based scores), and the binomial coefficient $\binom{k}{2}$ is the total number of possible pair combinations with k responsive stimuli. Analogously, we defined the mean association score for the other pairs $AS_{R-NR}$ as

$$AS_{R-NR} = \frac{1}{k \cdot (N-k)} \sum_{i=1}^{k} \sum_{j=k+1}^{N} a_{ij} \tag{3}$$

**Topographic organization.** For each electrode with at least two responsive units separated after spike sorting, we considered the degree of association between the images eliciting responses in the different units (of the same electrode). This way, we assessed if there is a topographic organization of the responses, with nearby neurons (that is, recorded from the same electrode) tending to fire to associated items. As before, we defined a mean association score between the images eliciting responses in the different neurons using equation (2), and compared it with the mean association score between the images eliciting responses in one of the neurons but not in the other one (equation (3)).

**Probability of pair responses.** The previous analyses were focused on neurons (and nearby neurons for the study of topographic organization) with more than one response, to assess whether these pairs of responses were to images that tend to be highly associated. We also evaluated the probability of neurons to respond to a pair of images, as a function of the degree of association between the images. But in this case we should as well consider neurons with single responses, because to evaluate the probability of neurons firing to two pictures with a given association score, we should consider not only the cases where neurons fired to both of them but also the cases where neurons fired to one (even if this was the neuron's unique response) and not the other one. Therefore, for this analysis we considered all 550 neurons (260 single units and 290 multi-units) with at least one response. To obtain a curve of probability of 'pair responses' as a function of degree of association, we binned the z-score-normalized web-association scores between all the presented images into 10 equally spaced bins, and we calculated, for each responsive neuron and bin, the ratio between the number of 'pair responses' and the number of responses to one but not the other image of the pair, and then averaged across neurons.

**Correlation between web-association and neuronal responses.** For each of the 550 responsive units we calculated a matrix of joint neural responses $F_{ij} = F_i \cdot F_j$, where $F_i$ and $F_j$ are the responses (that is, the median number of spikes in the

'response window'; see above) to pictures i and j, respectively, normalized by the maximum response. Note that for this analysis we did not use a criterion of what should, and what should not, be considered a significant response, but values below the neuron's mean baseline activity plus 2 spikes were capped to zero, to decrease the influence of background noise. This resulted in 151 units with a matrix F filled with zeros (that is, there were not two or more responses different from zero) and we therefore focus in the remaining 399 units (174 single- and 225 multi-units) with non-zero matrices (Supplementary Table 1). We also computed the web-association scores between all the pictures shown in the corresponding experimental session, and created the matrix of association scores A, where in this case we normalized each row using a z-score, given that the joint response matrix is non-zero for relatively few pairs and the fact that, in spite of the normalization of equation (1), some items (for example, a very famous actor) tended to have higher association scores than others. Then, for each neuron we computed the scalar product between the matrix of joint neural responses F and the matrix of web-association scores A:

$$r = \sum_{i=1}^{N} \sum_{j \neq i} A_{ij} F_{ij} \tag{4}$$

To assess the significance of the correlations between the neural and web-association matrices, for each neuron we statistically compared, using a rank test with $P < 0.05$, the original correlation value of equation (4) with the ones obtained from a distribution 1,000 surrogates, created by randomly permuting the index of the responses F. At the population level we compared the original correlation values with the median of the surrogates for each neuron using a right-sided (that is, our hypothesis was that the original values were higher than the ones of the surrogates) Wilcoxon signed-rank test.

The average joint-neural activation matrix (Supplementary Fig. 3) was obtained by averaging all the matrices of joint neural responses obtained from the 550 responsive neurons. Since not all the possible pairs of stimuli were shown together in at least one session, the matrix contains some holes corresponding to the untested pairs.

**Decoding analysis.** To predict the association scores, we took each non-zero entry $F_{ij}$ in the joint-neural response matrix and predicted the association score between the corresponding items (i,j) using a nearest-neighbour approach with a leave-one out validation (one at a time, each value was predicted based on the rest). That means, if we denote by $F'_{ij}$ the closest value to $F_{ij}$ in the joint-neural response matrix, and by $A'_{ij}$ its corresponding association score, we took $A'_{ij}$ as an estimation of $A_{ij}$. As before, the web-association matrices were normalized row by row and we used the mean of the symmetric values in the matrix, $\widehat{A_{ij}} = (A_{ij} + A_{ji})/2$ (for $i > j$), because otherwise the closest value would trivially tend to be the symmetric counterpart. Note that this analysis could only be applied to joint-neural response matrices with at least two entries different than zero, which was the case for 345 units (136 single- and 209 multi-units; Supplementary Table 1). We quantified the accuracy of the predictions with the mean square error,

$$\varepsilon = \frac{1}{K} \sum_{K} \left( \widehat{A_{ij}}' - \widehat{A_{ij}} \right)^2 \tag{5}$$

where $\widehat{A_{ij}}'$ and $\widehat{A_{ij}}$ are the predicted and real association scores, respectively, and K is the number of non-zero values in the joint-neural response matrix (with $i > j$). As before, for each neuron we used a rank test with $P < 0.05$ to compare the mean square errors of the original estimations with the ones obtained from a distribution 1,000 surrogates, created by randomly permuting the index of the responses F. At the population level, we compared the original errors with the median of the surrogates for each neuron using a left-sided (that is, our hypothesis was that the original errors were lower than the ones of the surrogates) Wilcoxon signed-rank test.

**Visual similarity.** In principle, it could be argued that the tendency of MTL neurons to respond to associated concepts could just reflect perceptual similarities between pictures (for example, a picture of an actor is more similar to a picture of another actor than to one of a landscape). To rule out this confound, we estimated the visual similarity $v_{ij}$ between each pair of stimuli i and j, as the cross-correlation between the images (each with $160 \times 160$ pixels, greyscale and z-score-normalized). Then, we calculated the tendency of neurons to respond to perceptually similar images, by replacing $a_{ij}$ by $v_{ij}$ in equations (2) and (3). The difference in the values between the response pairs (R-R) with the other pairs (R-NR) was significantly higher when considering the web-association scores compared with the visual similarity scores for the single units ($P < 0.005$; Wilcoxon signed-rank test; $n = 129$—differences for the multi-units were not significant), thus showing that results cannot be just attributed to perceptual similarity between the images. To further support this claim, we also repeated this analysis using the two-dimensional cross-correlation between the images and the Earth Mover's Distance[39], which are less sensitive to alignment issues. In both cases, the visual similarity metric gave significantly lower differences between the R-R and the R-NR values compared with the ones obtained with the web-based association scores ($P < 10^{-3}$ in all cases, both for the single- and the multi-units; Wilcoxon signed-rank test; $n = 129$ and 132).

**Effect of the relative number of hits.** Given that MTL neurons tend to respond to familiar people or places (for example, a very famous actor, landscape and so on)[18], it is in principle possible that our results could be explained by a higher probability of having responses (and joint responses) to very familiar items, irrespective of their degree of association. To rule out this confound, we calculated the product of the number of hits obtained for each independent search:

$$f_{ij} = \log_2\Big(\text{hits}(\text{concept}_i)\cdot\text{hits}(\text{concept}_j)\Big) \qquad (6)$$

Then, we replaced $a_{ij}$ by $f_{ij}$ in equations (2) and (3) and found that the scores for the pairs to which the neurons responded (R-R) were not significantly different to the ones for the other pairs (R-NR) ($P = 0.8$ for single units, $n = 129$ and $P = 0.4$ for multi-units, $n = 132$; Wilcoxon signed-rank test), thus showing that results cannot be explained by an effect introduced by the larger number of hits of familiar pictures.

As an alternative control, for each experimental session we divided the pictures of each category (actors, places, sportsmen and so on) into two groups: high familiarity (H; pictures with number of hits above the median) and low familiarity (L; pictures with number of hits below the median). We then calculated the association scores (equations (2) and (3)) for pair responses where pictures where both highly familiar (HH), both with low familiarity (LL) and one highly familiar and the other one not (HL). Replicating the results shown in Fig. 3b, for each of the three subgroups (HH, LL and HL) $AS_{R-R}$ was significantly higher than $AS_{R-NR}$ ($P < 0.05$ in all cases, both for single- and multi-units; Wilcoxon signed-rank test; $n = 64$, 63 and 93 for the single units, and $n = 72$, 71 and 106 for the multi-units). Moreover, the difference between the three subgroups was not significant ($P = 0.3$ for single- and 0.08 for multi-units; analysis of variance; $n = 220$ and 249), thus showing that our result cannot be attributed to the familiarity of the pictures used.

**Encoding of semantic categories.** To test that our results were not simply due to broad semantic category responses, we performed the following analyses. For pairs of items eliciting responses in a neuron that belonged to the same category (taking the categories of Fig. 2a), we compared the difference between the $A_{R-R}$ and $A_{R-NR}$ values of equations (2) and (3) but considering only association values within this category. The difference between $A_{R-R}$ and $A_{R-NR}$ remained significant for the single units ($P < 0.005$; Wilcoxon signed-rank test; $n = 103$) and was close to reaching significance for the multiunits ($P = 0.06$; Wilcoxon signed-rank test; $n = 116$). For pair of items eliciting responses that belonged to different categories (for example, an actor and a place), we constrained the comparisons to be across the same categories (for example, comparing the original association value with the ones between other actors and places). In this case, the difference between $A_{R-R}$ and $A_{R-NR}$ remained significant for the single units ($P < 0.01$; Wilcoxon signed-rank test; $n = 102$) and showed a tendency for the multi-units ($P = 0.09$; Wilcoxon signed-rank test; $n = 112$). These results show that the association coding described in the main text cannot be just attributed to semantic categorizations.

**Data availability.** The data that support the findings of this study are available from the corresponding author on reasonable request.

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

## Acknowledgements

We thank the patients for their participation. This work was supported by grants from the Human Frontiers Research Program.

## Author contributions

R.Q.Q., M.J.I., E.D.F. and I.F. designed the research; I.F. performed the surgeries; M.J.I. and R.Q.Q. collected and preprocessed the data; E.D.F. and M.J.I. performed the data analysis; E.D.F. and R.Q.Q. wrote the paper. All authors discussed the results and implications and commented on the manuscript.

**Additional information**

**Competing financial interests:** The authors declare no competing financial interests.

**How to cite this article**: De Falco, E. *et al.* Long-term coding of personal and universal associations underlying the memory web in the human brain. *Nat. Commun.* **7,** 13408 doi: 10.1038/ncomms13408 (2016).

