## [Peer Review File · Nature Communications]

Reviewers' comments:

Reviewer #1 (Remarks to the Author):

This is an exciting and valuable study on the firing properties of neurons in the medial temporal lobe of humans, characterizing lasting associative properties of neurons and thus suggesting that these memory representations do not disappear from the hippocampus following a consolidation period. The experimental design is clever and the analyses sophisticated and appropriate (if sometime hard to follow). The results are compelling and reveal in humans the existence of long lasting associative memory representations which parallels recent studies on place cells using a method of recording for long periods. I am enthusiastic about publication in Nature Communications but have some strong recommendations about the interpretation and comparison to data in animals.

1. They provide us with more specific localization of the electrodes within the medial temporal lobe. How many were recorded in each structures (e.g., hippocampus, amygdala)? Were there differences among these areas in associative coding?
2. They need to elaborate their discussion of the distinction between "association" and "category". Things that are in the same semantic category are often if not always associated - precisely by common category membership. Examples will help.

Reviewer #2 (Remarks to the Author):

The current study sought to examine the durability of associative representations in individual neurons in the human medial temporal lobe. To address this question, the authors identified neurons responsive to two or more stimuli and performed several analyses to determine whether these specific stimuli were associated with one another. Analysis of association scores between stimuli, rated either by the participants themselves or via a web-based approach, revealed that scores were higher when a neuron responded to both stimuli in the pair rather than a single stimulus. The authors also report a significant correlation between neuronal and stimulus association matrices, as well as the ability to predict the association score between two stimuli based on neuronal responses to these stimuli.

This is an interesting paper with the potential to broaden our understanding of the role of the human medial temporal lobe in associative memory. However, the paper could be strengthened and its novelty highlighted through consideration of the following points.

Major concerns:

1. For pairs of stimuli receiving high association scores, it seems likely that when participants view item A of the pair, pattern completion leads them to recall item B of the pair. As such, a neuron's response during the presentation of item A may reflect pattern completed recall of item B rather than neuronal tuning and newly preferential responding to item A itself. To give a more concrete example, imagine a neuron initially responsive only to

Prince William; after his wedding, the neuron also responds to Kate Middleton. It is possible that viewing a photo of Kate Middleton simply caused pattern-completed recall of Prince William, such that the apparent responsiveness to Kate Middleton instead reflects the same original response to Prince William. Can the authors rule out such an interpretation?

2. The authors perform careful control analyses to ensure that (a) neuronal and stimulus association scores are not driven solely by low-level perceptual similarities between images of the same category, and (b) their results hold when comparing associations within and between categories. I wonder, however, if the authors found any regional differences regarding responsiveness to certain categories of items. Given prior research demonstrating greater PHC responsiveness to scenes vs. faces, was it the case that PHC neurons responding to two or more stimuli typically responded to landmarks? And, given prior research demonstrating less content sensitivity in the hippocampus than MTL cortex, was it the case that the stimuli to which hippocampal neurons responded spanned these categories more equally? Finally, if multiple electrode locations were available in PHC within/across patients, was posterior PHC (i.e., closer to PPA) more responsive to landmarks than anterior PHC?

3. I'm not entirely sure that I understand what is being plotted in Figure 2B. Specifically, what do the individual points represent?

Minor points:

1. Minor grammatical mistakes can be found throughout the paper (e.g., "Using similar type of calculations...", "This result support a long term coding...", etc). Although each individual instance is minor, together they reduce the overall quality of the writing.

2. For Figure 1B, the numerical labels for the photos on the Y axis do not match those of the X axis or of Figure 1A.

Reviewer #3 (Remarks to the Author):

This manuscript clarifies how long-term memories may be supported by the responses properties of individual neurons in the MTL. The question is of very high interest to neuroscientist in the memory field and others. The manuscript presents data of great quality analyzed in a very robust way. The use of the internet joint searches as a proxy to test for links between stimuli and associated neural responses is an elegant cross-disciplinary method to probe the data. The statistical methods (surrogate data allowing computing significance and non-parametric tests), are appropriate and strongly support conclusions. The conclusions reached provide a great conceptual advance on the question of how memories/items associated through life are supported by individual cells. I recommend publication and my comments below are only there to help clarify the manuscript.

1) The statement in the intro "there is so far no direct evidence of neurons coding (or not)

previously acquired and not task related associations" falsely omit the work cited in ref 11 which shows that well learned associations are represented in the hippocampus through a greater selectivity. This does not make the present study less interesting as naturally formed associations and addresses a larger database. However, for accuracy, the work by Yanike should be appropriately cited.

2) The construction of the experiment is strangely laid out at bottom of intro line 46 to 50; it is difficult to understand whether the authors refer to this study or a previous one when they say "following an initial observation of neurons responding to well-known"

3) What are the categories of stimuli represented in the neuronal population? Is it possible to get a figure representing the nature of the responses across patients? From the examples, one may surmise that MTL neurons are particularly interested in sports. This may falsely generate the interpretation of the presence of semantic categorization.

Is it somehow possible to present an average activation map of the joint neural normalized responses for all the stimuli to see if overall some categories are more represented than others? This may be superposed on the "matrix of web-association scores for the pictures used in all the experimental sessions".

An average map would also give an idea about the diversity of the responses covered, and following the analogy with place cells that fire for all locations, show that cells may encode all categories.

4) Can the authors give a better account of the presence of responsive neurons across patients?

Were these neurons found in all patients? I understand that the proportion is in general above chance, but still, it would be good to know about its representation across patients? Can the authors also provide a clearer description of the database across regions across patients? Unless I am wrong, the only description is "Here we report data from sites in the hippocampus, amygdala, entorhinal cortex, and parahippocampal cortex."

This could be included in table S1 or alternatively in another table.

5) It is difficult to understand whether all responsive neurons could be used for the web metric analysis. I assume that the web metric analysis cannot be performed including the personally related stimuli (family members of the patient). If it is true that neurons are triggered by personal material more than other (viskontas, and figure 1), then, it might be possible that several neurons may not be used for the web based metrics because the stimuli would not be in the web (family members). Can the authors give some information about this, just for general comprehension and accuracy of representation of the database.

6) Iine 111. What does the term familiar represents exactly given that this paragraph is given in framework with ref 18 (Viskontas et al) . Is familiar referring to how often people are exposed to a stimulus (e.g. a picture of a sitcom that was watched everyday) vs a picture of an actor that is only seen in a movie per year? Or personally familiar stimuli as in ref 18? In ref 18, only HPC responded to familiar and personally familiar stimuli more than other. Other areas only show greater responses to personally familiar but not to familiar stimuli. So, the analysis might be relevant for HPC cells only, and not others as there is no familiarity effect.

I assume here that the analysis here only uses familiar as "I know who is in the picture

based on general knowledge, and not episodic personal experience".

Are the results the same if only hpc cells are used?

7) One puzzling finding is that multi-units tend to fire to the same pairs while neighboring neurons do not. The findings are paradoxical unless activity in the "multiunit" is driven by one main unit?

8) The result in the main text (line 108) relative to the control for visual similarity should be more explicitly referenced as this is a key control. The current wording leads to the false assumption that the conclusion is based on the previous findings of the team. (i.e. in agreement with ...we found that ...)

9) For accuracy the 2 following studies may be worth mentioning in the discussion

- Hampson et al. as previous evidence for incidental (not learned) coding of category related images by individual neurons without clear visual similarity (Hampson RE, Pons TP, Stanford TR, Deadwyler SA. Categorization in the monkey hippocampus: a possible mechanism for encoding information into memory. Proc Natl Acad Sci U S A. 2004 Mar 2;101(9):3184-9.)

- Reddy et al as evidence for coding incidentally learned associations (Reddy L, Poncet M, Self MW, Peters JC, Douw L, van Dellen E, Claus S, Reijneveld JC, Baayen JC, Roelfsema PR. Learning of anticipatory responses in single neurons of the human medial temporal lobe. Nat Commun. 2015 Oct 9;6:8556.).

Reviewer #4 (Remarks to the Author):

This work utilizes intracranial single and multi-unit recordings in the human medial temporal lobe to provide evidence that the hippocampus and related structures maintain long-term representations of associations between known concepts. Additionally, these representations were found to be distributed throughout neurons in a non-topographic manner. These are extremely interesting and novel findings and the analyses performed to reach these findings are well thought out and executed. Some issues need to be addressed:

1. Visual Similarity between pictures - Cross -correlation was used to estimate visual similarity between images. Was this a pixel by pixel correlation coefficient ($\frac{\sum(A_k * B_k)}{\sqrt{\sum(A_k^2) * \sum(B_k^2)}}$), A_k and B_k are the k th pixel in each image)? This would be sensitive to spatial alignment and orientation. The sum of a 2D cross correlation (Matlab: $\sum(\sum(xcorr2(A,B)))$) would be less sensitive to alignment. Another method would be to compare histograms of the pixel values with, for example, the Earth Mover's Distance. I don't expect the results to change much with a different image comparison method, and there's no one right way to do it, but a more established method or the same method with an explanation of why it was chosen is needed.

Rubner Y, et al. The Earth Mover's Distance as a Metric for Image Retrieval. (2000). Int J of Comp Vision. 40:2.

2. Image categories - Categories were manually assigned to broad categories (Actors, Musicians, Places, etc). The manually assigned categories are reasonable because the

categories are distinct. The authors mention in the methods that the categories were manually assigned only in the figure legend. It should be discussed in the body of the manuscript. The data in Figure 2a would benefit from a clustering analysis based on web-based associations. This would strengthen the findings of the paper.

3. The prediction results are weak. Is there any indication of what makes the small subset of units with predictions greater than chance different from the rest of the population of units? Are there generally more (or less) entries in the joint response matrix for these units? The authors have to provide more discussion about the results of the decoding analysis and a figure showing # of entries in joint response matrix vs prediction error.

Minor issues:

1. Information regarding units used for analyses - As human single-unit recordings and publications based on these recordings are still fairly rare, more information on this method would be helpful for readers to better understand the findings of this study. In particular, I would like to see:

a. Table S1 - Add a row for units that were recorded which did not fit any of the criteria for analyses

b. How many units were detected per micro-wire across all subjects (mean and std, and the actual distribution plotted on a histogram)?

c. How many units (and multi-units) fit the criteria in Table S1 per micro-wire, subregion, and participant?

2. Line 231 - Han Solo, not Hans Solo

Reviewer #1 (Remarks to the Author): This is an exciting and valuable study on the firing properties of neurons in the medial temporal lobe of humans, characterizing lasting associative properties of neurons and thus suggesting that these memory representations do not disappear from the hippocampus following a consolidation period. The experimental design is clever and the analyses sophisticated and appropriate (if sometime hard to follow). The results are compelling and reveal in humans the existence of long lasting associative memory representations which parallels recent studies on place cells using a method of recording for long periods. I am enthusiastic about publication in Nature Communications but have some strong recommendations about the interpretation and comparison to data in animals.

We thank the reviewer for the positive feedback and we have introduced her/his recommendations in the revised version.

1. They provide us with more specific localization of the electrodes within the medial temporal lobe. How many were recorded in each structures (e.g., hippocampus, amygdala)? Were there differences among these areas in associative coding?

In the revised version, we have introduced Table S2, which details the distribution of recorded (and responsive) neurons for each MTL area.

The differences in associative coding among areas are summarized in Figure 3D. Briefly, we saw a tendency for the association score to be different across MTL areas, being highest in Hippocampus and lowest in Parahippocampal cortex. This difference was close to significant but didn't reach the significance threshold ($p=0.07$, ANOVA). This is detailed in the section "Spatial organization" (in Results).

2. They need to elaborate their discussion of the distinction between "association" and "category". Things that are in the same semantic category are often if not always associated - precisely by common category membership. Examples will help.

As suggested by the reviewer, we have expanded the discussion about this issue in the second paragraph of the discussion. In particular, we agree with the reviewer that things that are in the same semantic category are not always associated and we now discuss examples showing that responses cannot be attributed to broad categories and reflect, instead, specific associations. At the population level, this observation is supported by the fact that: i) the association scores for items eliciting responses that belonged to the same category (e.g. two actors) were larger than the ones for other items of the same category (other actors), and ii) we also found responses across categories (e.g. to an actor and a place) and in this case, again, the association scores between these items was significantly larger than the ones for other items of the same

categories (other actors and places). These results are described in the last paragraph of the section “Web-based association metric” (in Results), they are discussed in the 2nd paragraph of the Discussion, and the details of the calculations are explained in the section “Encoding of semantic categories” (in Methods).

Reviewer #2 (Remarks to the Author):

The current study sought to examine the durability of associative representations in individual neurons in the human medial temporal lobe. To address this question, the authors identified neurons responsive to two or more stimuli and performed several analyses to determine whether these specific stimuli were associated with one another. Analysis of association scores between stimuli, rated either by the participants themselves or via a web-based approach, revealed that scores were higher when a neuron responded to both stimuli in the pair rather than a single stimulus. The authors also report a significant correlation between neuronal and stimulus association matrices, as well as the ability to predict the association score between two stimuli based on neuronal responses to these stimuli.

This is an interesting paper with the potential to broaden our understanding of the role of the human medial temporal lobe in associative memory. However, the paper could be strengthened and its novelty highlighted through consideration of the following points.

We thank the reviewer for the feedback and we address her/his comments below.

Major concerns:

1. For pairs of stimuli receiving high association scores, it seems likely that when participants view item A of the pair, pattern completion leads them to recall item B of the pair. As such, a neuron's response during the presentation of item A may reflect pattern completed recall of item B rather than neuronal tuning and newly preferential responding to item A itself. To give a more concrete example, imagine a neuron initially responsive only to Prince William; after his wedding, the neuron also responds to Kate Middleton. It is possible that viewing a photo of Kate Middleton simply caused pattern-completed recall of Prince William, such that the apparent responsiveness to Kate Middleton instead reflects the same original response to Prince William. Can the authors rule out such an interpretation?

The reviewer raises an important issue, namely, that the response to a given item A may reflect a recall of an associated item B. We consider, however, this interpretation very unlikely for two reasons. First, in these experiments subjects saw about 100 images per session, without doing any associative learning task. So, with such large number of images it seems unlikely that the subjects will spontaneously and consistently recall specific associations evoked by the picture presentations. Moreover, note that the neurons fired to some but not all associated items (as shown in Figure 3e and in the exemplary neurons shown). This is in strong contrast to other paradigms where subjects have to learn specific associations between very few pictures (as in one of our

previous works: Ison et al, Neuron 2015), in which case it is indeed more likely that some responses may be due to the recall of an associated item.

Second, to further rule out the reviewer's concern we calculated the latency of the responses and compared them with the latencies obtained in a previous cued-recall study (also with single cell human MTL recordings) reported by Kreiman and colleagues (Kreiman et al.,2000). Kreiman et al. report that the mean latency of human MTL responses given by evoking an image from a cue was of 409ms (s.d.: 291ms), which was about 130ms longer than the one they found for the visual responses (triggered by the picture presentations). In our case, using the same latency estimation method of Kreiman et al, the mean latency of the responses was 253 ms (s.d.: 129 ms), which is well below the one reported by Kreiman et al for cued-recall responses. Moreover, the mean latency difference of the responses to different stimuli (in the same neurons) was of 78 ms, whereas the latency difference reported by Kreiman et al for recall responses was more than 60% larger.

2. The authors perform careful control analyses to ensure that (a) neuronal and stimulus association scores are not driven solely by low-level perceptual similarities between images of the same category, and (b) their results hold when comparing associations within and between categories. I wonder, however, if the authors found any regional differences regarding responsiveness to certain categories of items. Given prior research demonstrating greater PHC responsiveness to scenes vs. faces, was it the case that PHC neurons responding to two or more stimuli typically responded to landmarks? And, given prior research demonstrating less content sensitivity in the hippocampus than MTL cortex, was it the case that the stimuli to which hippocampal neurons responded spanned these categories more equally? Finally, if multiple electrode locations were available in PHC within/across patients, was posterior PHC (i.e., closer to PPA) more responsive to landmarks than anterior PHC?

In the revised version of the paper we have introduced Figure S2b, which shows the proportion of responses to each category, for each MTL area. As noted by the reviewer, there is a larger proportion of responses to scenes in PHC compared to other areas. In agreement with this observation, there was a significant modulation of the proportion of responses to scenes across the different MTL areas (ANOVA, $p < 10^{-6}$). No effect was found for any of the other categories. We would prefer to report this result elsewhere, as we currently have a paper under revision in which we examine in detail these PHC scene responses (not only showing that PHC neurons tend to fire more to scenes, but also analysing the tuning within this category, to what type of scenes they tend to fire most, etc). However, within the context of this paper, it is important to rule out that our results are not a trivial consequence of such category responses in PHC. For this, we have repeated the analysis but excluding responses to scenes and have found that our main claim holds, namely, that pairs of pictures that MTL neurons fire to tend to be associated ($p < 10^{-7}$ when considering all pictures and $p < 10^{-5}$ when excluding scenes).

In response to the reviewer's question, we found no significant difference in the proportion of responses to the different categories of stimuli in the hippocampus (ANOVA, $p=0.7$). Finally, it would certainly be interesting to compare the specific category of the responses across different PHC locations, but unfortunately we don't have the possibility to address this point, as we cannot provide such a specific localization of the micro-wires with which we record the neurons' activity (we can broadly identify the area where they are placed but since they are a brush protruding from the depth electrode tip we cannot localize them more precisely).

3. I'm not entirely sure that I understand what is being plotted in Figure 2B. Specifically, what do the individual points represent?

Figure 2b shows the correlation between the personal association scores (assigned by the patients) and the web-based association scores. For this, we considered the web-based association scores for which we also had the personal score given by the patient; then we binned these web-based scores into 10 equally spaced intervals, and plotted the mean (\pm SEM) value in each bin (x-axis) against the mean (\pm SEM) value of the corresponding personal scores (y-axis). We have clarified this in the Methods section (subsection "Correlation between the personal and the web-based association metrics").

Minor points:

1. Minor grammatical mistakes can be found throughout the paper (e.g., "Using similar type of calculations...", "This result support a long term coding...", etc). Although each individual instance is minor, together they reduce the overall quality of the writing.

We have fixed these mistakes and have double-checked the article to fix grammatical errors.

2. For Figure 1B, the numerical labels for the photos on the Y axis do not match those of the X axis or of Figure 1A.

Thanks for noticing this error, which has been fixed.

Reviewer #3 (Remarks to the Author):

This manuscript clarifies how long-term memories may be supported by the responses properties of individual neurons in the MTL. The question is of very high interest to neuroscientist in the memory field and others. The manuscript presents data of great quality analyzed in a very robust way. The use of the internet joint searches as a proxy to test for links between stimuli and associated neural responses is an elegant cross-disciplinary method to probe the data. The statistical methods (surrogate data allowing computing significance and non-parametric tests), are appropriate and strongly support conclusions. The conclusions reached provide a great conceptual advance on the question of how memories/items associated through life are supported by individual cells. I recommend publication and my comments below are only there to help clarify the manuscript.

We thank the reviewer for the positive feedback and the suggestions for improvement.

1) The statement in the intro "there is so far no direct evidence of neurons coding (or not) previously acquired and not task related associations" falsely omit the work cited in ref 11 which shows that well learned associations are represented in the hippocampus through a greater selectivity. This does not make the present study less interesting as naturally formed associations and addresses a larger database. However, for accuracy, the work by Yanike should be appropriately cited.

In this point we politely disagree with the reviewer. Note that in the discussion section we do mention that the work of Yanike and colleagues (as well as the one from Miyashita et al, and others cited in the discussion) deals with well-learned associations: "Closer to our study, selective responses to well-learned associations were described in the monkey ¹¹ and the rat hippocampus ¹²." The sentence of the introduction refers, however, to "previously acquired and not task related associations", which we believe is correct because in Yanike et al responses were obtained while the animals performed a learning task, whereas in our case responses to associated pictures were obtained during passive viewing (i.e. thus showing a long term coding irrespective of having subjects doing an associative-learning task).

2) The construction of the experiment is strangely laid out at bottom of intro line 46 to 50; it is difficult to understand whether the authors refer to this study or a previous one when they say "following an initial observation of neurons responding to well-known"

We have clarified this in the revised version, where we now say: "...following the previous observation of neurons responding to well known and allegedly associated concepts (e.g. two co-stars in a TV show) ^{16,17}..."

3) What are the categories of stimuli represented in the neuronal population? Is it possible to get a figure representing the nature of the responses across patients? From the examples, one may surmise that MTL neurons are particularly interested in sports. This may falsely generate the interpretation of the presence of semantic categorization.

Is it somehow possible to present an average activation map of the joint neural normalized responses for all the stimuli to see if overall some categories are more represented than others? This may be superposed on the "matrix of web-association scores for the pictures used in all the experimental sessions".

An average map would also give an idea about the diversity of the responses covered, and following the analogy with place cells that fire for all locations, show that cells may encode all categories.

Following the reviewer's suggestion, we have added Figure S2c, showing the distribution of responsive stimuli across patients and categories.

Moreover, as suggested by the reviewer, in the new version of the paper we have added Figure S3, which shows the neural activation map for all stimuli (i.e. the average joint-neural responses considering all 550 responsive neurons). Although the map contains many holes (as not all the possible pairs of stimuli have been tested together), it shows the diversity of responses spanning all categories. Most importantly, using the same quantification described in the section "Correlation between the web-association matrices and the neurons' responses" (see Figure 4), we found that there was a significant correlation between this average joint-neural matrix of responses and the matrix of web-association scores (Figure 2a). We have added this result in the revised version ("Cell-by-cell and decoding analysis" section).

4) Can the authors give a better account of the presence of responsive neurons across patients?

Were these neurons found in all patients? I understand that the proportion is in general above chance, but still, it would be good to know about its representation across patients? Can the authors also provide a clearer description of the database across regions across patients? Unless I am wrong, the only description is "Here we report data from sites in the hippocampus, amygdala, entorhinal cortex, and parahippocampal cortex."

This could be included in table S1 or alternatively in another table.

In the revised version we have included Figure S1, which shows the distribution of responsive neurons across patients and areas, as suggested by the reviewer. In the figure it can be observed that, in spite some expected variability, responsive neurons were present across the different patients and MTL areas. Also following the reviewer's suggestion, the number of recorded (and responsive) neurons in the different MTL areas is now shown in Table S2.

5) It is difficult to understand whether all responsive neurons could be used for the web metric analysis. I assume that the web metric analysis cannot be performed including the personally related stimuli (family members of the patient). If it is true that neurons are triggered by personal material more than other (viskontas, and figure 1), than, it might be possible that several neurons may not be used for the web based metrics because the stimuli would not be in the web (family members). Can the authors give some information about this, just for general comprehension and accuracy of representation of the database.

The reviewer is correct that personal stimuli are usually more likely to trigger responses in MTL neurons and that these responses couldn't be used in the web metric analysis. We have clarified this in the revised version of the methods section ("Web-based association metric").

6) line 111. What does the term familiar represents exactly given that this paragraph is given in framework with ref 18 (Viskontas et al) . Is familiar referring to how often people are exposed to a stimulus (e.g. a picture of a sitcom that was watched everyday) vs a picture of an actor that is only seen in a movie per year? Or personally familiar stimuli as in ref 18? In ref 18, only HPC responded to familiar and personally familiar stimuli more than other. Other areas only show greater responses to personally familiar but not to familiar stimuli. So, the analysis might be relevant for HPC cells only, and not others as there is no familiarity effect.

I assume here that the analysis here only uses familiar as "I know who is in the picture based on general knowledge, and not episodic personal experience". Are the results the same if only hpc cells are used?

In the passage the reviewer refers to, we talk about familiarity in a general sense. As the reviewer said, in this sense familiar people are those that the subjects are more often exposed to, and since we took the number of hits in an internet search as a measure of familiarity, the analysis we use is indeed based on general knowledge and not on the personal experiences by each subject. The important point, following ref 18, is that familiar pictures are represented by more neurons and it could, in principle, be argued that the finding of neurons encoding associations between these pictures is by chance. This is, however, not the case as we show with the joint familiarity analysis.

To address the reviewer's final question, we have also run the joint familiarity analysis (same as detailed in the Method section 'Effect of the relative number of hits') for each area separately. We found that the result holds (no significant effect of familiarity) when considering only responses in the hippocampus (signed-rank test, $p=0.6$ for single-units, $p=0.9$ for multi-units), as well as when considering each of the other MTL areas separately ($p>0.05$ in all cases) .

7) One puzzling finding is that multi-units tend to fire to the same pairs while

neighboring neurons do not. The findings are paradoxical unless activity in the "multiunit" is driven by one main unit?

The finding that multi-units fire to associated concepts could indeed appear paradoxical, if one assumes that their responses to different stimuli come from different neurons. This is, however, not necessarily the case. In fact, as the reviewer noted, we believe that the responses found in a given multiunit are in most cases attributable to one single unit and thus the finding of firing to associated concepts. In other cases, though, the responses of the multiunit may be given by more than one nearby neuron and we have therefore obtained an average lower association score for the multi-units compared to the single-units.

8) The result in the main text (line 108) relative to the control for visual similarity should be more explicitly referenced as this is a key control. The current wording leads to the false assumption that the conclusion is based on the previous findings of the team. (i.e. in agreement with ...we found that ...)

We agree with the referee and have edited the text to clarify this point.

9) For accuracy the 2 following studies may be worth mentioning in the discussion

- Hampson et al. as previous evidence for incidental (not learned) coding of category related images by individual neurons without clear visual similarity (Hampson RE, Pons TP, Stanford TR, Deadwyler SA. Categorization in the monkey hippocampus: a possible mechanism for encoding information into memory. Proc Natl Acad Sci U S A. 2004 Mar 2;101(9):3184-9.)

- Reddy et al as evidence for coding incidentally learned associations (Reddy L, Poncet M, Self MW, Peters JC, Douw L, van Dellen E, Claus S, Reijneveld JC, Baayen JC, Roelfsema PR. Learning of anticipatory responses in single neurons of the human medial temporal lobe. Nat Commun. 2015 Oct 9;6:8556.)

Thanks for the references, which are now mentioned in the discussion.

Reviewer #4 (Remarks to the Author):

This work utilizes intracranial single and multi-unit recordings in the human medial temporal lobe to provide evidence that the hippocampus and related structures maintain long-term representations of associations between known concepts. Additionally, these representations were found to be distributed throughout neurons in a non-topographic manner. These are extremely interesting and novel findings and the analyses performed to reach these findings are well thought out and executed. Some issues need to be addressed:

We thank the reviewer for the positive feedback and we address her/his comments below.

1. Visual Similarity between pictures - Cross -correlation was used to estimate visual similarity between images. Was this a pixel by pixel correlation coefficient ($\frac{\sum(A_k * B_k)}{\sqrt{\sum(A_k^2) * \sum(B_k^2)}}$, A_k and B_k are the k th pixel in each image)? This would be sensitive to spatial alignment and orientation. The sum of a 2D cross correlation (Matlab: $\sum(\sum(xcorr2(A,B)))$) would be less sensitive to alignment. Another method would be to compare histograms of the pixel values with, for example, the Earth Mover's Distance. I don't expect the results to change much with a different image comparison method, and there's no one right way to do it, but a more established method or the same method with an explanation of why it was chosen is needed.

Rubner Y, et al. The Earth Mover's Distance as a Metric for Image Retrieval. (2000). Int J of Comp Vision. 40:2.

Yes, we did employ a pixel by pixel correlation on the grayscale versions of the stimuli. Following the reviewer's suggestion, we repeated this analysis using the sum of the 2D cross correlation and the Earth Mover's Distance and as she/he expected, results remained the same: in both cases, the visual similarity metric gave significantly lower differences between the R-R and the R-NR visual similarity scores compared to the ones obtained with the web-based association scores ($p < 10^{-3}$ in all cases, both for the single- and the multi-units). We have added this result in the revised version of the paper (in the section: "Visual Similarity").

2. Image categories - Categories were manually assigned to broad categories (Actors, Musicians, Places, etc). The manually assigned categories are reasonable because the categories are distinct. The authors mention in the methods that the categories were manually assigned only in the figure legend. It should be discussed in the body of the manuscript. The data in Figure 2a would benefit from a clustering analysis based on web-based associations. This would strengthen the findings of the paper.

As suggested by the reviewer, we have moved this information to the main text (the fact that labels were assigned manually is now mentioned in the Results and in the Methods sections). Also following the reviewer's suggestion we have done a clustering analysis, using the Matlab implementation (Rubinov and Sporn, 2008) of the Louvain community detection algorithm (Blondel et al., 2008), which gave a 84% overlap between the manual classification and the one given by the clustering algorithm. This is now mentioned in Results ("Web-based association metric" section) and Methods ("Web-based association metric" section).

3. The prediction results are weak. Is there any indication of what makes the small subset of units with predictions greater than chance different from the rest of the population of units? Are there generally more (or less) entries in the joint response matrix for these units? The authors have to provide more discussion about the results of the decoding analysis and a figure showing # of entries in joint response matrix vs prediction error.

As we argue in the article, the main reason for having a relatively weak prediction (illustrated with the examples of Figures S8 and S9) is due to the fact that MTL neurons encode relatively few of the associations between the pictures presented in each session. Consequently, we expect to have a correlation between the number of entries in the joint response matrix and the prediction error, as the reviewer suggested. This was indeed the case, as shown in the figure below plotting the prediction error (minus the prediction error of the 5-percentile of the surrogates) vs the number of non-zero responses (giving the number of entries in the joint response matrix). In fact, we observe that the largest errors are obtained when having few non-zero entries. Overall, the correlation was significant with $p < 0.005$. In the corrected version we describe this result at the end of the section "Cell-by-cell and decoding analysis"

Minor issues:

1. Information regarding units used for analyses - As human single-unit recordings and publications based on these recordings are still fairly rare, more information on this method would be helpful for readers to better understand the findings of this study. In particular, I would like to see:

a. Table S1 - Add a row for units that were recorded which did not fit any of the criteria for analyses.

Done

b. How many units were detected per micro-wire across all subjects (mean and std, and the actual distribution plotted on a histogram)?

The mean number of detected units per micro-wire was 1.6 (s.d.: 0.8). This is now reported in the Methods section (“Spike sorting and responsiveness criteria”). The figure below shows the distribution of detected units per microwire:

c. How many units (and multi-units) fit the criteria in Table S1 per micro-wire, subregion, and participant?

In the revised version we have added Table S2 and Figures S1 and S2 showing the distribution of units per participant, per MTL subregion and also per category of response.

2. Line 231 - Han Solo, not Hans Solo

Indeed! Thanks for spotting the typo.

REVIEWERS' COMMENTS:

Reviewer #1 (Remarks to the Author):

The authors have directly and adequately addressed my concerns.

Reviewer #2 (Remarks to the Author):

The authors were very responsive to all four reviewers' concerns, and the manuscript is stronger as a result. In particular, the addition of new analyses and results (especially those that are category-related) provides greater support for the authors' interpretation of the data. Together with the clarification of various methodological points, these revisions allow the authors to present a much-improved manuscript. That said, I believe that the manuscript could be further strengthened through consideration of the following point:

In their response letter, the authors' provided a helpful response to the question of whether the reported results could simply reflect pattern completion rather than true associative coding. Given that many readers from the memory community may have the same concern, I suggest that the authors include their response in the manuscript (or supplement) itself. In particular, inclusion of the response latency analysis would be particularly useful in attenuating concerns about a pattern completion explanation, the latter of which is not as novel a finding as an associative coding interpretation.

Reviewer #3 (Remarks to the Author):

I have no further recommendation and thank the authors for the clarification in the main text and the addition of the supplementary material. Although it appears tedious, the supplementary documenting the responses across patients shows the robustness of the phenomenon the authors are describing, and that their effect is not driven by responses observed in a few patients.

I vividly recommend publication and give my best wishes to the authors to keep doing their exciting research.

Reviewer #4 (Remarks to the Author):

The authors have adequately addressed all my concerns.